# Can Waist-to-Height Ratio and Health Literacy Be Used in Primary Care for Prioritizing Further Assessment of People at T2DM Risk?

**DOI:** 10.3390/ijerph20166606

**Published:** 2023-08-18

**Authors:** Elín Arnardóttir, Árún K. Sigurðardóttir, Marit Graue, Beate-Christin Hope Kolltveit, Timothy Skinner

**Affiliations:** 1School of Health, Business and Natural Sciences, University of Akureyri, 600 Akureyri, Iceland; 2Health Care Institution of North Iceland, 580 Siglufjordur, Iceland; 3Akureyri Hospital, 600 Akureyri, Iceland; 4Department of Health and Caring Sciences, Western Norway University of Applied Sciences, 5063 Bergen, Norway; 5Institute of Psychology, University of Copenhagen, 1017 Copenhagen K, Denmark; 6Australian Centre for Behavioral Research in Diabetes, Melbourne, VIC 3053, Australia

**Keywords:** prediabetes, countryside/town, screening, well-being, type 2 diabetes

## Abstract

Background: To identify people at risk of type 2 diabetes. Primary health care needs efficient and noninvasive screening tools to detect individuals in need of follow-up to promote health and well-being. Previous research has shown people with lower levels of health literacy and/or well-being scores are vulnerable but may benefit from intervention and follow-up care. Aims: This cross-sectional study, aimed to identify people at risk for type 2 diabetes by comparing the Finnish Diabetes Risk instrument with the waist-to-height ratio. Further, the difference was examined in health literacy and well-being scale scores in the countryside versus town areas, respectively. Results: In total, 220, aged 18–75 years, participated. Thereof, 13.2% displayed biomarkers at prediabetes level of HbA1c (39–47 mmol/mol); none had undiagnosed diabetes. Of the participants, 73% were overweight or obese. Waist-to-height ratio demonstrated 93.1% of the prediabetes group at moderate to high health risk and 64.4% of the normal group, with an area under the curve of 0.759, sensitivity of 93.3%, and specificity of 63.1%. Residency did not influence prediabetes prevalence, health literacy, or well-being. Conclusion: Waist-to-height ratio and the Finnish Diabetes Risk instrument may be suitable for identifying who need further tests and follow-up care for health promotion in primary care.

## 1. Introduction

Type 2 Diabetes Mellitus (T2DM) risk or prediabetes biomarkers indicate elevated risk for the individual, contributing to the development of insulin resistance in the outlining of T2DM disease [1]. Prediabetes is defined by elevated levels of HbA1c, impaired fasting glucose (IFG), or impaired glucose tolerance (IGT), which is based on a 2 h oral glucose tolerant test (OGTT), above the normal range but not reaching the diagnostic level of T2DM [2].

Prediabetes biomarkers are linked to an increased risk of up to a ratio of 1:2 progression to T2DM and an increased risk for several serious comorbid conditions such as cardiovascular disease (CVD), which may be in progression before T2DM diagnosis [3]. Additionally, up to one-third of people with a body mass index (BMI) indicating obesity have prediabetes signs [1]. It is estimated that the interval between the onset and diagnosis of T2DM is reported to be up to 7 years in the US [4], with nearly half of cases unaware of their T2DM condition [5]. Current projections are that, by 2030, 478 million people worldwide will have T2DM [6]. A crude estimation of US adult prediabetes prevalence in 2017–2020 was 38% [7]. In Iceland, the prevalence of T2DM has been increasing, in 2018 prevalence in Iceland was 3.5% in women and 4.1% in men showing similarity in prevalence and incidence as of US 20 years earlier. According to the Icelandic Heart Association population study, a total of 10,600 individuals were diagnosed with T2DM in 2018 [8]. By comparing the medical prescription database and T2DM prevalence data, an underestimation of 29% in T2DM prevalence was established [8].

Population studies in Iceland show a high obesity prevalence [9], estimated to be 20% in 2007 and 27% in 2017 [8]. The prevalence of undiagnosed T2DM is uncertain in Iceland, but research indicates that, in northern Iceland, prediabetes prevalence, here as based on the HbA1c diagnostic level of American Diabetes Associations (ADA), may be 13.2% [10]. If the increasing levels of diabetes are to be stemmed, it is important to develop simple and effective screening strategies to identify those who are the most at risk, for instance, those with obesity or prediabetes.

Although biomedical tests such as HbA1c, 2 h OGTT and fasting glucose can be used for diagnostic purposes, these are not always readily implementable across sparsely populated countries, such as Iceland [2]. However, questionnaires like the Finnish Diabetes Risk Score (FINDRISC) have been designed to estimate the risk of developing T2DM over the next 10 years [11]. In addition, measurements ratios, including BMI for obesity, waist-to-height Ratio (WHtR), and waist-to-hip Ratio (WHR) have been used for overall, diabetes, or cardiovascular health risk assessments [12,13].

In addition to these traditional biometric measures, poorer mental health and well-being has been associated with an increased risk of prediabetes and progression to T2DM [14]. Persons with T2DM may also have double the likelihood of depression than in the general society [15]. Thus, measuring well-being, health-related quality of life (HRQoL), and depression signs may have additional benefits in identifying those with prediabetes and undiagnosed T2DM [16].

Health literacy (HL) is the capability to gather, understand, judge, and follow complex information and demands of what may be good for one’s health to prevent disease and promote health [17], that is, motivating one to improve quality of life [17,18,19]. Better HL positively influences health outcomes [19]. A systematic review of limited HL prevalence in T2DM points to the fact that limited HL may affect self-management and empowerment and that may lead to poorer health outcomes for people with T2DM [20]. The status of HL between countries may vary because several factors may influence HL in addressing the interactive, critical, or functional level of HL [21]. However, the variance in HL may assist the addressing and decision-making of integrated healthcare services and follow-up aiming to improve HL for individuals [20].

Furthermore, HL has been found to be a key in self-management and in personalizing services for T2DM patients [21,22], correlating to lower health literacy and poorer wellbeing and worse T2DM management [18,20,22].

Primary healthcare (PHC) needs to address if and which noninvasive screening may identify people at T2DM risk, before further and more invasive testing and categorizing within the risk group are conducted. Also, if HL or well-being show any characteristics within risk groups. Giving the PHC the opportunity to concentrate first on high-risk individuals while providing less intrusive methods for low-risk individuals [23].

This cross-sectional study aimed to find the best suitable and noninvasive methods for identifying people at risk of T2DM, by using HbA1c measurement in comparing the sensitivity and specificity of FINDRISC scale scoring with WHtR, BMI, and WHR. Furthermore, the present study has examined if differences were found at the HL level and well-being scale scores among people living in northern Iceland, that is, in the countryside versus town. In addition, the present study has explored if HL and/or well-being scale scores may contribute to identifying vulnerable groups regarding T2DM risk, and who are in need of targeted support.

## 2. Materials and Methods

This was a cross-sectional study inviting participants at risk of diabetes but not diagnosed with diabetes to three of the largest Primary Health Clinics in North Iceland. The study was conducted as a pre-phase of an intervention study that followed. The research was launched at the early beginning of the COVID-19 pandemic in Iceland. Data collection was completed via one-on-one interviews with the first author between 1 March 2020 and 15 May 2021.

### 2.1. Participants and Data Collection

All inhabitants were eligible for participation if they were (a) aged 18–75 years and living in the service area of the three PHCs of the Health Institution of North Iceland included in the research, (b) not diagnosed with diabetes, and (c) spoke and understood Icelandic or English fluently.

The original plan was to recruit participants via introduction letters handed out by a receptionist to all patients fulfilling age criteria who visited the participating PHC clinics. But the authorities placed strict restrictions on all visits to PHC clinics both for the public and all ‘outsiders’ (researchers) when the COVID pandemic hit, resulting in changes from the original plan of approaching participants visiting the PHC clinics.

Almost half of the participants (n = 101) were recruited, as originally planned, from the end of February until the beginning of May 2020. The remaining (n = 119) participants were recruited via flyers and advertisements in local papers from January 2021 to May 2021.

The data were collected by the first author (EA) at the PHC clinics or at the research center when the PHC clinics were locked down because of COVID restrictions. We collected data on height, weight, waist, and haemoglobin A1c protein (HbAlc) (see Section 2.3 and Section 2.3.1). In addition, the participants answered questionnaires to collect data on age, gender, educational level, working and living status, family history of T2DM, and whether they had been diagnosed with diabetes. In addition, question 7 in the FINDRISC instrument asks if an individual has ever been diagnosed with high glucose levels, including when pregnant. The participants also filled in diabetes risk, health literacy, and well-being questionaries (see Section 2.3.2, Section 2.3.3, Section 2.3.4 and Section 2.3.5).

### 2.2. Geographical Area and Layout of Primary Health Care in Iceland

Iceland is divided into seven health districts; in each, there is a health institution coordinating small area hospitals and PHC clinics, that are financed by the state. PHC locations have historical backgrounds, based on distribution of settlements, and challenging traveling in winter [24,25]. The northern district has scattered agricultural areas beside industry, service, and fishing settlements, with an increasing focus on tourism over the past decade [25]. Around 36,000 of the total 368,792 inhabitants of the country (as of 1 January 2021) [26] were living in northern Iceland, with around 17,600 in the service area of the three PHCs participating in the study. Location was Akureyri the town area and Husavik and Saudarkrokur the countryside areas, each with one sub-rural town. According to Statistic Iceland definition of population density in Iceland, that are based on international definitions. The terms town and countryside describe the areas better than the terms urban and rural according to density of inhabitants and service provided in the areas included in the study [26].

In Iceland, especially in northern Iceland, the winters are both cold and dark, and even though there are paved main roads, they are often filled with ice and snow. However, improved transportation in recent decades and the internet have significantly reduced isolation in more dispersed localities [26]. Though public transportation is limited, private cars are common [26]. Cultural activities and infrastructure are historically strong in smaller, more rural towns and in the surrounding areas in Iceland, with access to education such as high school, industrial, and vocational training [26].

### 2.3. Biological Measurements, Demographic Definitions, and Instruments

Measurements of height to the nearest 0.1 cm with a portable measuring tape and weight in light clothing on a digital scale to the nearest 100 g calculating BMI as kg/m^2^—in which overweight was determined as >25 kg/m^2^ and obesity was determined as ≥30 kg/m^2^—were taken [27]. Also, waist (2 cm over navel) and hip (at widest point) measurements, here with 1.5 or 3 m capacity measuring tape and evaluating consistency of the measuring tape regularly, enabled calculation of both WHR and WHtR. Using a definition from the World Health Organization (WHO) for Europe, WHR caused increased a health risk for men at ≥0.94 and ≥0.80 for women [28]. Former research indicates better predictive power for the waist-to-height ratio than BMI for diabetes risk, defining a WHtR of <0.5 as no increased health risk, 0.5–0.6 as increased to high health risk, and ≥0.6 as very high health risk [29].

#### 2.3.1. Haemoglobin A1c Protein (HbA1c) Measurements, Diabetes and Prediabetes Definition Levels

The HbA1c level, which is an approved measurement as a diagnostic test of T2DM by WHO [2], was analysed with capillary blood samples by a ‘DCA Vantage^®®^’(Siemens Medical Solutions Diagnostics Europe Limited, Dublin, Ireland). Here, the ADA classification was used, which define prediabetes stage as HbA1c levels between 39 and 47 mmol/mol (5.7–6.4%) and levels of ≥48 mmol/mol (≥6.5%) as having T2DM [30].

#### 2.3.2. Finnish Diabetes Risk Score (FINDRISC) Instrument

FINDRISC was developed in Finland as a risk scale of T2DM in the next 10 years and is an approved diagnostic test of diabetes risk [31]. It is easy to use, inexpensive, non-invasive, and validated in several populations [32], including in Iceland [11]. It is scored on a scale from 0 to 26 points, with a higher score representing higher T2DM risk [31].

To identify those at risk, the reported best cut-off points may vary between countries, from 11 points in Bulgaria [33] to 15 points in Norway [34], where lower income status in the country may have an influence towards the lower cut-off point [35]. Here, using ≥11 points gave the best sensitivity and specificity for finding those at HbA1c level of prediabetes, with area under the curve (aROC) at 0.814 (CI 0.733–0.896) [10].

#### 2.3.3. Health Literacy (HL) Questionnaire

The Icelandic version of the European Health Literacy questionnaire (HLS-EU-Q16) was used [19]. The HLS-EU-Q16 has been explicitly described before [36]. The Icelandic HLS-EU-Q16-IS consists of 16 questions on a four-answer scale ranging from ‘very difficult’ and ‘fairly difficult’ (either giving 0 points) to ‘fairly easy’ and ‘very easy’ (either giving 1 point). Scores are summarized, with the final score from 0 to 16 points; a higher score represents better HL, which is categorized into inadequate HL (scoring 0–8), problematic HL (scoring 9–12), and sufficient HL (scoring 13–16). Valid HLS-EU-Q16IS responses include no more than two missing questions [19].

#### 2.3.4. World Health Organization Well-Being Index (WHO-5)

The WHO-5 questionnaire measures subjective psychological well-being with five questions, that ask the respondent about their well-being in the previous two weeks. Each question is answered on a scale from 0 to 5, giving a maximum of 25 points. Multiplying the raw scoring of WHO-5 by four gives a range of well-being on a scale from 0 (absent well-being) to 100 (maximal well-being). A score of <50 is an indicator of reduced psychological well-being and a score of ≤28 is an indicator of depression [37]. It has been validated in many languages and countries [37], including Iceland [38].

#### 2.3.5. Europe Quality of Life Five Dimension Five Level Instrument (EQ-5D-5L)

The EQ-5D-5L is a standardized tool for health status measurement [39], that enables the measurement of HRQoL [40]. The instrument contains two parts: a descriptive component with five dimensions (5D) of mobility, self-Care, usual activities, pain/discomfort, and anxiety/depression. The dimensions are responded to on five levels (5L): (1) ‘no’, (2) ‘slight’, (3) ‘moderate’ (4) ‘serve’, and (5) ‘unable/extreme’ problems, with a total of 3125 plausible results [39]. Results of each dimension are reported as a code. The code 11111 gives information of ‘health state’ of answering no problems of all dimensions, but the code 11155 a ‘health state’ of no problem with the first three dimensions but unable/extreme problems regarding pain/discomfort and anxiety/depression. The second part of the EQ-5D-5L includes a Visual Analogue Scale (EQ-VAS) that measures self-reported health status on a scale from 0 (the worst possible health you can image) to 100 (the best health you can image) [39,40].

### 2.4. Statistical Analysis

Descriptive statistics were used describing continuous variables to calculate means, standard deviations, and ranges. Relative risk and odd ratio were calculated using crosstabs. For categorial variables, counts and proportions were used. The sample characteristics, according to residency or groups of normal HbA1c levels versus prediabetes levels of HbA1c, were calculated by independent t-tests for continuous variables, chi-square tests for categorical variables, and retested with ANOVA or nonparametric chi-square tests when appropriate.

Microsoft Excel was used to calculate the BMI, WHR, and WHtR of each participant. Chi-square tests were used in calculations for comparison of WHtR and WHR in finding people at risk of T2DM; this was carried out by dividing the sample into two groups of the normal HbA1c group and prediabetes level group. Normal HbA1c or prediabetes level were used to compare the accuracy of sensitivity and specificity calculation to find people at prediabetes biomarkers with the cut off score of ≥11 points on FINDRISC [10] and the same aROC calculation of WHtR, BMI, and WHR. A prefect accuracy of aROC with neither false positives nor false negatives is 1, but 0.5 indicates the results are no better than chance. The best cut-off points for identifying people with prediabetes biomarkers were the shortest distance to the upper left corner of the ROC curve [41].

When comparing variables according to residency, results were controlled for age. Correlation and regression calculations were used for estimation of the background variable relationships to HLS-EU-Q16-IS and WHO-5 questionnaires. EQ-5D-5L is reported as the health state index of each of the five health dimensions from 1 to 5. A value set for calculations of Quality of Adjusted life Years (QUALY) is not yet available for Iceland [39].

The dataset was analysed using IBM SPSS statistics 27. If applicable, missing data were excluded listwise. Significant statistical difference (two tailed) was *p* ≤ 0.05.

### 2.5. Ethical Considerations

The present study was performed in accordance with the Helsinki Declaration and with the approval of the Icelandic National Bioethics Committee (VSN), (VSN-19-080-V1 approved 14 January 2020. All participants received and read an informational letter and signed an informed consent form before participating.

Trial registration: This study is a pre-phase of the registered study ‘Effectiveness of Nurse-coordinated Follow-Up Programme in Primary Care for People at Risk of T2DM’ at www.ClinicalTrials.gov (NCT01688359) (accessed on 30 December 2020).

## 3. Results

### 3.1. Main Findings

The majority of participants reported daily exercise and had no family history of T2DM. But 13.2% were found with HbA1c biomarkers of prediabetes, none with undiagnosed diabetes. When controlled for age, neither residency nor gender influenced prevalence of prediabetes biomarkers. BMI levels of overweight and obesity were high. People with increased overall health risk according to WHtR had 7.463 grater odds of having HbA1c biomarkers of prediabetes. WHtR found 68.2% participants at overall increased health risk when FINDRISC, using a cut-off point of ≥11 points, found 39.1% to be at increased diabetes risk. Residency had no influence on well-being, but being a man, age, and prediabetes biomarkers showed correlation to higher score on WHO-5. Health literacy and well-being questionaries gave added information not included in the FINDRISC instrument. Findings will now be described in more details.

### 3.2. Characteristics of the Study Participants According to Residency

A total of 220 individuals participated, of which 66% were female. The background information is presented in Table 1. There was an equal gender distribution between residencies. Countryside residents were significantly older than town residents *p* < 0.001. The educational level was high, but town residents had a higher educational level *p* < 0.05.

### 3.3. Biological Measurements and Results from FINDRISC

The results of biological measurements and score on FINDRISC are shown in Table 2. No individuals were found to have undiagnosed T2DM and 13.2% of the participants had an HbA1c level indicative of prediabetes.

On FINDRISC, 92.3% reported daily exercise. Those found with HbA1c prediabetes biomarkers, were less likely to exercise (t_(218)_ = 2.07, *p* = 0.04 (two tailed)). No family history of diabetes was reported by 62.7%, 21.4% had T2DM history by second relatives and 15.9% by first relatives. Supported by participants with HbA1c biomarker levels of prediabetes and family history, the majority had no family history of diabetes (62.1%), but 34.5% had first relatives with diabetes.

FINDRISC scores were significantly lower for the normal HbA1c group than the prediabetes group, at 8.6 (SD ± 4.5) and 14.7 (SD ± 5.2), respectively (*p* < 0.001). Using a cut-off point of ≥11 points on FINDRISC gave a sensitivity of 79.3% and a specificity of 67%, showing 86 participants at increased T2DM risk, thereof 63 with normal HbA1c levels and 23 at prediabetes HbA1c levels.

BMI ranged from 18.5 to 48.2 kg/m^2^. The results showed that no participants were underweight, but 78.4% of the countryside and 67.9% town residents were overweight or obese (see Table 2).

The normal HbA1c group had a lower BMI, at *M* = 28.3 (SD ± 5.2), than the prediabetes biomarker HbA1 group *M* = 32.3 (SD ± 5.7); *t*_(218)_ = −3.618, *p* < 0.001. An aROC of the BMI calculation using HbA1c as a definition of prediabetes gave a result of an aROC of 0.713, *p* < 0.001 (CI 0.624–0.803) sensitivity of 96.6% and specificity of 69.1% at a BMI of 25.0 kg/m^2^.

Differences in health risk evaluation were found between countryside and town residents when using either the WHR or WHtR (see Table 2). Of the prediabetes biomarkers group, 89.7% were in the higher health risk WHR group, but this was also true for 62.3% of the normal HbA1c group. WHtR measurements found 93.1% of the HbA1c prediabetes biomarkers group at a moderate to high health risk and 64.4% of the normal HbA1c group.

An aROC curve calculation using low or high health risk according to the results of WHR to find people at prediabetes biomarker levels of HbA1c showed the aROC to be 0.654 (CI 0.563–0.745) with a sensitivity of 93.1% and a specificity of 62.3% (*p* = 0.008). For WHtR, 0.5 was used as a point of increased health risk, with an aROC of 0.759 (CI 0.668–0.851), *p* < 0.001, a sensitivity of 93.3% and a specificity of 63.1%.

The odds ratio of having HbA1c biomarkers of prediabetes was found to be 7.46 times greater for those with high risk WHtR value than those of low risk, (95% confidence interval 1.72 to 32.35), *p* = 0.002.

Looking at the 59 participants with normal HbA1c levels, who scored ≥11 points on FINDRISC and had a BMI > 25 kg/m^2^, the WHR risk grouping identified 54 as having higher health risk. However, WHtR identified all 59 as having an overall higher health risk. Analyses of WHtR and HbA1c results according to gender showed that 92.9% of men and 93.3% of women with HbA1c prediabetes biomarkers had a WHtR indicating a higher overall health risk.

### 3.4. Health Literacy and Wellbeing Instruments

Table 3 reports the results of HLS-EU-16IS, EQ-5D-5L, EQ-VAS, and WHO-5 instruments, finding no significant difference according to residency. Of the 220 participants, 211 fulfilled the requirements of the HLS-EU-16IS and were included in the HL results. The majority (83.4%) scored sufficient HL. Some participants paused when answering the HL instrument reporting that some items in the HL instrument did not apply to them because they had never been in the situation presented in Q3: ‘Understanding what your doctor says to you’, Q5: ‘Judge when you may need to get a second opinion from another doctor’ and/or Q11: ‘Judge if the information on health risk in the media is reliable’. Some then said out loud ‘well that would probably not be a problem’.

The WHO-5 result found a significantly higher mean score *M* = 72.71 (SD ± 24.4) for the prediabetes biomarker group than the normal HbA1c group *M* = 62.1 (SD ± 25.8), *t*_(206)_ = −2.035; *p* = 0.043. Men scored significantly higher than women on WHO-5 *M* = 68.8 (SD ± 25.0) and *M* = 60.7 (SD ± 25.9), respectively, (*t*_(206)_ = 2.161, *p* = 0.032). Positive correlation was found between the scoring of WHO-5 to age *r*_(208)_ = 0.273 (CI 0.142–0.39) *p* < 0.001.

Scoring on the EQ-5D-5L showed that most defined themselves as having no problems (Level 1) with mobility (84.1%), self-care (96.4%), and usual activities (87.7%). Only 31.8% did so for pain/discomfort and 60% for anxiety/depression; 41 reported levels 3 to 5 in pain/discomfort and 25 did so for anxiety/depression. Interestingly, there was no significant correlation between the dimensions of mobility and anxiety/depression (*p* = 0.125 two-tailed) which was also true for the dimensions of self-care and anxiety/depression, (*p* = 0.991 two-tailed). Other dimensions showed correlation between each other at the *p* < 0.001 level (two-tailed). There was a negative correlation between anxiety/depression and the total score on the WHO-5 Well-Being index; a lower reported level of anxiety/depression correlated to a lower score on WHO-5 well-being (*p* < 0.001).

In 14.1% of the answers, the score on EQ-VAS was <70. There was correlation of the EQ-VAS scoring to all dimensions of EQ-5D-5L (*p* < 0.001). On EQ-VAS scoring, neither residency (*p* = 0.320), gender (*p* = 0.726), nor HbA1c at ADA level of prediabetes (*p* = 0.255) influenced the EQ-VAS score.

## 4. Discussion

This study used different screening methods to identify people at risk of T2DM in PHC clinics in need of follow-up to promote health and well-being. Residency did not influence the results. Interestingly, with aROC = 0.814 for FINDRISC, 82 of the 86 with ≥11 points on FINDRISC were at increased overall health risk based on WHtR, in addition 68 participants were at increased risk according to WHtR but scored < 11 points on FINDRISC. Therefore, WHtR ratio found more people at an overall higher health risk, and for this cohort, it might suggest a plausible underestimation of future T2DM risk in the next 10 years when using only the FINDRISC.

WHtR of ≥0.5, gave lower aROC = 0.759 than FINDRISC, but nearly 7.6 greater odds ratio of having HbA1c biomarkers of prediabetes. Distinguishing better those at prediabetes biomarker risk than BMI or WHR (aROC = 0.713 and 0.654, respectively). Supported by earlier research, indicating WHtR to be a better overall health risk measurement, especially for women [42]. In addition, the results indicate that WHtR may be more suitable than WHR or BMI for identifying individuals at risk of prediabetes level, T2DM disease, and/or CVD [43,44]. In a systematic review, WHtR has been found to show increased health risk at 0.5 for adults, children, and different ethnic groups and there is high specificity and sensitivity for WHtR outcome measurements of T2DM and CVD risk [13]. Though both WHR and BMI have been found to be predictors of T2DM and CVD risk, they have different criteria for both gender and ethnic groups and need more calculations than WHtR [13].

The prevalence of overweight (39.1%) and obesity (34.1%) was higher than the OECD country health profile of 2021 reporting obesity to be 27% in 2017 in Iceland [45]. Nearly half (47.4%) of the overweight or obese scored ≥11 points on FINDRISC. The PHC must respond to this as research show satisfactory results of interventions within the PHC without medications helping people to reduce weight [46,47].

Prediabetes prevalence in the cohort was 13.2% according to HbA1c biomarkers alone [10]. Because the HbA1c indicates the glycation of red blood cells for the last two to three months using only one measurement of HbA1c, as was carried out here, we may have missed out individuals at T2DM risk [48]. It has been argued that this is why HbA1c cannot alone predict further development towards T2DM; rather, it gives indications for the need of further follow-up [48]. Although using the cut-off point of ≤38 mmol/mol has allegedly been said to exclude prediabetes [49], when individuals present risk through WHtR and FINDRISC results of >11 points, normal HbA1c will not exclude T2DM risk as found here. Research has also indicated that caution is needed when using HbA1c alone as a diagnostic tool to find people with T2DM because it may miss people at the IGT stage [48].

The PHC challenge is to select the most appropriate, simplest and accurate, non-invasive measurements and instruments finding individuals at T2DM risk, prioritizing assistance to high-risk individuals towards health promotion [12]. Therefore, before considering further and more invasive and costly tests and interventions, our suggestions are to use WHtR in PHC for the first measurement as an indicator of overall health risk and in screening for prediabetes and T2DM risk. Then, for the second measurement the FINDRISC (with ≥11 points marker for Iceland), should be added, followed by HbA1c, fastening blood glucose, and then OGTT measurement of those screened at higher T2DM risk.

### 4.1. Adding Health Literacy and Well-Being Questionnaires into the Screening Equation

High mean score of HLS-EU-Q16IS, with 83.4% having sufficient HL was not surprising considering the participants’ high educational level. Better HL has been associated with higher education, but lower HL with poorer health and quality of life [17].

Some participants reported that some items in the HL instrument did not apply to them because they had never been in the situation presented in Q3, Q5, and/or Q11. It remains unclear if this affected their responses in grading themselves with a higher HL, because, some said, ‘Well that would probably not be a problem’. However, scoring low on Q3 (‘Understanding what your doctor says to you’) had a correlation with increased health risk of WHtR, supported by research where 19% of adults with prediabetes presented low HL, scoring worse on questions on; ‘understanding health care professionals’, ‘difficulty in obtaining information’, and ‘understanding written information’ [50]. Adding the HLS-EU-Q16IS questionnaire to a non-invasive screening may assist in prioritizing an educational intervention for people at T2DM risk or higher health risk within the PHC.

Well-being scores increased with age and men scored significantly higher than women. Uneven gender proportion, self-selected participation and COVID-19 might have affected the results. Results from Denmark indicate that COVID-19 had a greater negative effect on WHO-5 scores for women than men [51]. In Iceland, depression is more prevalent among women (9%) than men (6.3%) [26]. We are unable to explain why participants with prediabetes biomarkers presented higher total scores on WHO-5, indicating fewer signs of depression. This contrasts a systematic review showing higher prevalence of lower well-being and depression in people with T2DM [52]. Neither depression nor anxiety symptoms are addressed in FINDRISC, but newly published results demonstrate that people with depressive or anxiety symptoms had a higher likelihood of T2DM [53]. The WHO-5 questionnaire might, from this perspective, assist the PHC in categorizing who at T2DM risk needs to be prioritized for further interventions.

The results of EQ-5D-5L of self-reported HRQoL showed over 8 out of 10 participants scored themselves with no problems on the first three levels: ‘Mobility’ (84.1%), ‘Self-care’ (96.4%), and ‘Usual activities’ (87.7%). Only one third reported no problems for ‘Pain/Discomfort’ and 60% for ‘Anxiety/Depression’, with no difference according to gender or prediabetes biomarkers. A systematic review on the EQ-5D-5L found higher utility value scores for men than women with T2DM [54]. This is in line with our WHO-5 results, of anxiety/depression with 10.5% scoring <28 points. Which is higher anxiety/depression rate than the 7.7% found in the Icelandic population in 2019 [26].

### 4.2. Plausible Effects of the Characteristics of the Participant’s Backgrounds

High Gross Domestic Product (GDP) of Iceland, accessible low cost PHC and high informational accessibility though common internet access [55], might explain no differences in HL and well-being according to residency. 

The uneven gender distribution, favouring women, needs to be addressed if it influenced the results, as generally, more men are diagnosed with diabetes [8] and in the general population of the areas, the gender distribution is near equal [25]. If T2DM risk prevalence in the next 10 years are equal to the results, the prevalence of T2DM may greatly increase in the coming decades. It is therefore important that the PHC finds those at risk to reverse the progression to T2DM in the future.

## 5. Conclusions

Results revealed that WHtR and FINDRISC seem to be effective and useful non-invasive measurements identifying people at T2DM risk. Starting with WHtR calculations in the PHC may categorize in advance those at higher health and T2DM risk from those at lower overall health risk. Also, three questions in the HLS-EU-Q16IS and the WHO-5 instrument were found helpful in categorizing further who might be in need of intervention. In PHC, the approach of using the simple WHtR measurement before more invasive and expensive testing methods can therefore be recommended.

### 5.1. Limitation

It is a limitation to this study that the study was conducted during the COVID-19 pandemic. Thus, the recruitment of patients from partly locked down clinics was more difficult. In addition, we gained an uneven gender distribution. Moreover, the anxiety measurements might not reflect the overall anxiety in the population because of the ongoing COVID pandemic and its impacts on mental well-being. It would have strengthened the data to have more than one measurement of HbA1c.

### 5.2. Strength of the Study

All measurements by one researcher.

Sample relatively large compared to the population.

### 5.3. What this Paper Adds

Prediabetes prevalence in North Iceland is 13.2% according to HbA1c biomarkers, which is lower than expected.

Surprisingly, residency did not influence well-being and the HbA1c prediabetes biomarker group reported higher well-being.

High BMI may call for turning to alternative measurements like WHtR that better identifies those at higher health risk than BMI.

Using the third question on HLS-EUQ16IS and WHtR, in addition to a FINDRISC score of ≥11, and HbA1c measurements may distinguish those needing further follow-up due to increased risk of developing T2DM.

### 5.4. What Is Already Known on This Subject

Prediabetes prevalence is uncertain as prevalence of T2DM is rapidly increasing worldwide. Iceland is now around 20 years behind the US in T2DM prevalence.

HbA1c has been criticized as sole biomarker for prediabetes identification as it may miss out individuals at high T2DM risk.

WHtR is an easily appliable measurement, showing increased early health risk for all, if results are over 0.5.

Primary health care needs simple, non-invasive, and non-expensive methods for identifying people at increased T2DM risk in order to turn the evolution of T2DM backward.

## Figures and Tables

**Table 1 ijerph-20-06606-t001:** Background characteristics of countryside vs. town participants.

	Countryside (n = 111)	Town (n = 109)	*p* Countryside/Town
Mean age (in years, 18–75 years)	55.3 (SD ±13.2)	48.9 (SD ±14.3)	*p* < 0.001 *
Age	n (%)	n (%)	
<45 years	19 (17.1)	35 (32.1)	
45–54 years	29 (26.1)	30 (27.5)	
55–64 years	31 (27.9)	25 (22.9)	
65 and over	32 (28.8)	19 (17.4)	
Gender	n (%)	n (%)	*p* = 0.602 **
Male	36 (32.4)	39 (35.8)	
Female	75 (67.6)	70 (64.2)	
Living status	n (%)	n (%)	*p* = 0.784 **
Alone	7 (6.3)	10 (9.2)	
With one other person	55 (49.5)	46 (42.2)	
With two or more persons	49 (44.1)	53 (48.6)	
Educational level	n (%)	n (%)	*p* = 0.049 **
Elementary/junior high or equal	31 (27.9)	21 (19.3)	
Upper secondary/vocational training/ Senior high school or equal	35 (31.5)	30 (27.5)	
University degree	44 (39.7)	57 (52.3)	
Educational level missing	1 (0.9)	1 (0.9)	
Occupational status	n (%)	n (%)	*p* = 0.632 **
Working partly or full time	84 (75.7)	81 (74.3)	
Unemployed	2 (1.8)	4 (3.7)	
Pensioner (disabled/elderly)	20 (18.0)	13 (11.9)	
Other ***/did not answer	5 (4.5)	11 (10.1)	

* Independent *t*-test. ** Chi-square test. *** Participant who marked multiple of the other three groups; one did not answer occupational status.

**Table 2 ijerph-20-06606-t002:** HbA1c levels, FINDRISC score, BMI, WHtR, and WHR according to residency.

	Defined as	Countryside (n = 111)	Town (n = 109)	*p*-Value Countryside/Town
HbA1c levels		n (%)	n (%)	
Mean (SD)		34.3 (SD ± 3.4)	35.3 (SD ± 4.0)	*p* = 0.048 *
24–38 mmol/mol	Normal	100 (90.1)	91 (83.5)	
39–47 mmol/mol	Prediabetes	11 (9.9)	18 (16.5)	
FINDRISC score		n (%)	n (%)	
Mean (SD)		10.1 (SD ± 4.5)	8.8 (SD ± 5.5)	*p* = 0.056 *
<11 points		62 (55.9)	72 (66.1)	*p* = 0.121 ^¥^
≥11 points		49 (44.1)	37 (33.9)	
BMI kg/m^2^		n (%)	n (%)	
Mean (SD)		29.5 (SD ± 5.5)	28.1 (SD ± 5.2)	*p* = 0.053 *
18–24.99	Normal	24 (21.6)	35 (32.1)	
25–29.99	Overweight	46 (41.4)	40 (36.7)	
30–39.99	Obese	33 (29.7)	31 (28.4)	
40>	Serve obese	8 (7.2)	3 (2.8)	
WHtR		n (%)	n (%)	*p* < 0.001 ^¥^
<0.5	No increased risk	22 (19.8)	48 (44.0)	
≥0.5 and <0.6	Increased to high risk	59 (53.2)	33 (30.3)	
≥0.6	Very high risk	30 (27.0)	28 (25.7)	
WHR		n (%)	n (%)	*p* < 0.001 ^¥^
♂ < 0.94 ♀ < 0.80	Low health risk	24 (21.6)	50 (45.9)	
♂ ≥ 0.94 ♀ ≥ 0.80	Higher health risk	87 (78.4)	59 (54.1)	

* Independent *t*-test, ^¥^ Chi-square test, ♂ men, ♀ women.

**Table 3 ijerph-20-06606-t003:** Scoring of the HLS-EU-Q16IS, the EQ-5D-5L, the EQ-VAS scale, and the WHO-5 instruments according to residency.

The HL-Q16IS Instrument	Countryside (n = 111) n (%)	Town (n = 109) n (%)	*p* Value ^¥^
Mean (SD)	14.5 (SD ± 2.3)	14.8 (SD ± 1.7)	0.276
Sufficient HL (13–16 points)	83 (74.8)	93 (85.4)	
Problematic HL (9–12 points)	20 (18.0)	13 (11.9)	
Inadequate HL (0–8 points)	1 (0.9)	1 (0.9)	
Missing/Insufficient answers	7 (6.3)	2 (1.8)	
WHO-5	n (%)	n (%)	
Mean (SD)	66.2 (SD ± 24.7)	60.9 (SD ± 26.7)	0.140
<08 total points	10 (9.0)	13 (11.9)	
28–49 total points	18 (16.2)	24 (22.1)	
50–100 total points	78 (70.3)	65 (59.6)	
Missing	5 (4.5)	7 (6.4)	
The EQ-5D-5L instrument	n (%)	n (%)	
Health state			
11111	22 (19.8)	36 (33.0)	
11112	7 (6.3)	4 (3.7)	
11121	26 (23.4)	20 (18.3)	
11122	19 (17.1)	14 12.8)	
11123	3 (2.7)	6 (5.5)	
11131	5 (4.5)	4 (3.7)	
21121	4 (3.6)	2 (1.8)	
All other (49 groups)	25 (22.5)	23 (21.1)	
EQ-VAS scoring 0–100	n (%)	n (%)	
Mean (SD)	81.0 (SD ± 17.9)	83.2 (SD ± 14.8)	0.320
<70	19 (17.1)	12 (11.0)	
70–89	45 (40.5)	40 (36.7)	
90–100	47 (42.3)	57 (52.3)	

^¥^ Independent *t*-test.

## Data Availability

The datasets used and/or analysed during the current study are available from the corresponding author on reasonable request.

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
