# Peer review of "Can Waist-to-Height Ratio and Health Literacy Be Used in Primary Care for Prioritizing Further Assessment of People at T2DM Risk?"

_ijerph, 2023, doi:10.3390/ijerph20166606_

Round 1
Reviewer 1 Report
This small study from Iceland showed that WHtR added to the prediction of people at high risk of T2D compared with the FINDRISC. This is interesting since the FINDRISC includes both BMI and waist in the model.
Only HbA1c was used in the study and as the authors point out this will miss many people having high glucose, fasting, or post-challenge.
That WHtR >.5 detected more people at high risk than the FINDRISC >11 points. However, this is obvious since more people had WHtR >.5 than FINDRISC >11 points. Thus the sensitivity of FINDRISC >11 points was lower due to the higher cut-point. This should be mentioned in the Discussion.
There were no significant differences in T2D risk levels between urban and rural areas when age was taken into account. Thus Tables showing various variables by urban/rural status are unnecessary. Also, I wonder whether the “urban” area in this part of the country was truly urban by international definitions.
The Discussion is too long. In many places, it can be shortened.
Author Response
Author’s Notes to Reviewer 1
Manuscript ID: ijerph-2482640: “Can Waist-to Height ratio and Health literacy be used in Primary Care for prioritizing further assessment of people at T2DM risk?
Thank you for reviewing our manuscript for publication in IJERPH and for constructive comments. Below are the comments from the reviewer 1 followed by our responses marked in red. We have answered all questions and revised the manuscript accordingly by using red text.
Remarks and responses:
- Appropriate research design: Can be improved.
Thank you for this comment. We have revised the text in the Materials and methods section to clarify, see lines 101-103, 112-114, and 136-143.
- Adequately described methods: Can be improved.
Thank you for this comment. We have taken 3 steps to improve the Methods section:
Firstly: we have added a paragraph in section 2.1 to improve the data collection description, see lines 112-129.
Secondly: in section 2.2 we have replaced the terms “urban or rural” with international definitions of “town and countryside “. We refer to information from Statistics Iceland to further describe the population density in Iceland, see lines 136-143.
Thirdly: section 2.4 has been rewritten to clarify the statistical analysis see lines 209-211 and 226.
- Clearly presented results: Must be improved.
Thank you for this comment. We have made the result section more precise and deleted redundant information from the tables as well as calculations of trends. In addition, we have added prevalence of daily exercise, family history of diabetes to the results, odds ratio of HbA1c and WHtR as suggested, see lines 242-253, 263-271 and 296-303. We have rewritten and shortened the discussion section.
- Conclusions supported by the results: Can be improved.
Thank you for this comment. We have improved the text as requested, lines 424-430.
- Comments and Suggestions for Authors:
- a) “This small study from Iceland showed that WHtR added to the prediction of people at high risk of T2D compared with the FINDRISC. This is interesting since the FINDRISC includes both BMI and waist in the model”.
Thank you for this comment. Waist-to-height ratio showed higher sensitivity and specificity compared to both BMI and waist hip ratio, however FINDRISC had the highest sensitivity and specificity with cut-off point of 11 points. Therefore, in the manuscript we recommend using both the WHtR and FINDRISC. But as the WHtR is simpler to use and does not require different calculations for gender, age, or ethnic groups it is recommended to start with that measurement.
- b) “Only HbA1c was used in the study and as the authors point out this will miss many people having high glucose fasting or post-challenge”:
Thank you also for this comment. That is why we found it necessary to address it in the discussion section, see lines 361-370.
- c) “That WHtR > .5 detected more people at high risk than the FINDRISC >11 points. However, this is obvious since more people had WHtR > .5 than FINDRISC > 11 points was lower due to the higher cut-point. This should be mentioned in the Discussion”.
Thank you for pointing this out. We have made changes in the discussion section to address this, see lines 345-355.
- d) “There were no significant differences in T2D risk levels between urban and rural areas when age was taken into account. Thus, Tables showing various variables by urban/rural status are unnecessary. Also, I wonder whether the “urban” area in this part of the country was truly urban by international definitions”.
Thank you for this comment. We have now revised the analysis presenting the data in terms of town and countryside. The terms town and countryside are according to the definition of Statistic Iceland. The terms town and countryside describe the differences between density of inhabitants and service provided in the areas of north Iceland, included in the study. The findings from this study adds to previous research as most research in Iceland until now have been conducted around the capital area of Reykjavík in the south. It is important to explore differences between the largest town of North Iceland to smaller countryside areas of North Iceland.
- e) “The Discussion is too long. In many places, it can be shortened”.
Thank you for this comment. We have reorganized and shortened the discussion.
On behalf of the authors,
Elin Arnardóttir
elin@unak.is
Reviewer 2 Report
This study by Arnardóttir et al. aims to tests the utility of non-invasive methods like waist-height ratio and health awareness to predict T2D risk in a set of 220 Icelanders. The work is intriguing; however, the study raises several questions/ concerns.
1. It seems all participants were non-diabetic. What criteria was used to diagnose non-diabetic status of the participants? HbA1c only? Fasting glucose? Single occasion HbA1c measures cannot be used as a primary criterion to differentiate diabetic and non-diabetic individuals. HbA1c readings are informative in presence of other glycemic measures, or if they are repeated at least 2 times. Did the authors do any?
2. Did the participants have any family history of T2D?
3. Was the difference in physical activity considered in all the analyses?
4. Did you analyze males and females separately? Especially the waist-height ratio and prediabetes status?
5. Authors do acknowledge that the study was done during COVID, however their anxiety measurements may be biased because there was extremely high prevalence of anxiety due to COVID pandemic and their measurements might not reflect the overall anxiety in the population but may be capturing the COVID related anxiety.
6. Prediabetes group showed a better well-being than normal group. Why do you think that would be? Are they more health-aware because of their condition?
7. It would be more informative to show the odds ratios and 95% confidence intervals of HbA1c related to waist-height ratio.
8. The sample size is small to determine the effectiveness of WHtR on T2D risk. It would have been nice to include some T2D patients in the study to capture an overall impact on T2D risk.
Author Response
Author’s notes to Reviewer 2
Manuscript ID: ijerph-2482640: “Can Waist-to Height ratio and Health literacy be used in Primary Care for prioritizing further assessment of people at T2DM risk?
Thank you for reviewing our manuscript for publication in IJERPH and for constructive comments. Below are the comments from the reviewer 1 followed by our responses marked in red. We have answered all questions and revised the manuscript accordingly by using red text.
Remarks and responses:
- Appropriate research design: Must be improved.
Thank you for this comment. We have revised the text in the Materials and methods section to clarify, see lines 101-103, 112-114, and 136-143.
- Adequately described methods: Must be improved.
Thank you for this comment. We have taken 3 steps to improve the Methods section:
Firstly: we have added a paragraph in section 2.1 to improve the data collection description, see lines 112-129.
Secondly: in section 2.2 we have replaced the terms “urban or rural” with international definitions of “town and countryside “. We refer to information from Statistics Iceland to further describe the population density in Iceland, see lines 136-143.
Thirdly: section 2.4 has been rewritten to clarify the statistical analysis see lines 209-211 and 226.
- Clearly presented results: Must be improved.
Thank you for this comment. We have made the result section more precise and deleted redundant information from the tables as well as calculations of trends. In addition, we have added prevalence of daily exercise, family history of diabetes to the results, odds ratio of HbA1c and WHtR as suggested, see lines 242-253, 263-271 and 296-303. We have rewritten and shortened the discussion section.
- Conclusions supported by the results: Must be improved.
Thank you for this comment. We have improved the text as requested, lines 424-430.
- It seems all participants were non-diabetic. What criteria was used to diagnose non-diabetic status of the participants? HbA1c only? Fasting glucose? Single occasion HbA1c measurements cannot be used as primary criterion to differentiate diabetic and non-diabetic individuals. HbA1c readings are informative in presence of other glycemic measurements, or if they are repeated at least 2 times. Did the authors do any?
We used HbA1c measurements to identify people with diabetes. This measurement was only performed once. We have added that information in the limitation section, lines 432-437. In addition, we used questionnaires for self-report (i.e., if they had been diagnosed with diabetes by a doctor, and again question 7 in the FINDRISC instrument asking if the individual had ever been diagnosed with high glucose levels, including when pregnant) see lines 126-129.
- Did the participants have any family history of T2D?
We have collected this information by the participants response on question 8 in the FINDRISC instrument. We have added this information in the Methods section and Results section, see lines 126-128 and 266-271.
- Was the difference in physical activity considered in all the analyses?
We have added the results of daily exercise in the result section, lines 243 and 266-267. The data showed that 92.3% of the participants reported daily exercise, and that the prediabetes group were less likely to exercise p = .04. As over 90% reported to be physical active this variable was not included in the statistical analyses.
- Did you analyze males and females separately? Especially the waist-height ratio and prediabetes status?
Yes, we did. We have added this information in the Results section, see lines 301-303.
- Authors do acknowledge that the study was done during COVID, however their anxiety measurements may be biased because there was extremely high prevalence of anxiety due to COVID pandemic and their measurements might not reflect the overall anxiety in the population but may be capturing the COVID related anxiety.
Thank you for pointing this out. We have added information from Statistic Iceland on the prevalence of depression for people 15 years and older in Iceland from 2019 in the discussion section, see lines 397-398 and 402-405. In addition, we have added that the anxiety measurements might not reflect the overall anxiety in the population because of the ongoing COVID pandemic and its impacts on mental well-being, lines 432-437.
- Prediabetes group showed a better well-being than normal group. Why do you think that would be? Are they more health-aware because of their condition?
We found these results surprising and contradictory to previous results. We have added some reflections on this matter to the discussion, see lines 398-405.
- It would be more informative to show the odds ratios and 95% Confidence intervals of HbA1c related to waist-height ratio.
We have added this information to the paper, see lines 296-298.
- The sample size is small to determine the effectiveness of WHtR on T2D risk. It would have been nice to include some T2D patients in the study to capture an overall impact on T2D risk.
We agree. However, the ongoing Covid-19 pandemic made it difficult to requite participants. We have included this in the limitation section, lines 432-437. Still, our sample is relatively large compared to the total population in the area.
On behalf of the authors,
Elin Arnardóttir
elin@unak.is

Round 2
Reviewer 2 Report
Authors addressed all my concerns.